# A Machine Learning Approach to the Diagnosis of Autism Spectrum Disorder and Multi-Systemic Developmental Disorder Based on Retrospective Data and ADOS-2 Score

**DOI:** 10.3390/brainsci13060883

**Published:** 2023-05-31

**Authors:** Marilena Briguglio, Laura Turriziani, Arianna Currò, Antonella Gagliano, Gabriella Di Rosa, Daniela Caccamo, Alessandro Tonacci, Sebastiano Gangemi

**Affiliations:** 1Unit of Child Neurology and Psychiatry, Department of Human Pathology of the Adult and Developmental Age “Gaetano Barresi”, Polyclinic Hospital University, 98125 Messina, Italy; marilena.briguglio@polime.it (M.B.); laura.turriziani@polime.it (L.T.); arianna.curro@polime.it (A.C.); antonella.gagliano1@unime.it (A.G.); 2Department of Biomedical Sciences, Dental Sciences and Morpho-Functional Imaging, Polyclinic Hospital University, 98125 Messina, Italy; 3Clinical Physiology Institute, National Research Council of Italy (IFC-CNR), 56124 Pisa, Italy; alessandro.tonacci@cnr.it; 4Unit of Allergy and Clinical Immunology, Department of Clinical and Experimental Medicine, Polyclinic Hospital University, 98125 Messina, Italy; sebastiano.gangemi@unime.it

**Keywords:** autism spectrum disorder, multisystem developmental disorder, retrospective data, ADOS2 score, machine learning, regression models, RIDGE model, Linear Regression Model, LASSO model, CART model

## Abstract

Early and accurate diagnosis of autism spectrum disorders (ASD) and tailored therapeutic interventions can improve prognosis. ADOS-2 is a standardized test for ASD diagnosis. However, owing to ASD heterogeneity, the presence of false positives remains a challenge for clinicians. In this study, retrospective data from patients with ASD and multi-systemic developmental disorder (MSDD), a term used to describe children under the age of 3 with impaired communication but with strong emotional attachments, were tested by machine learning (ML) models to assess the best predictors of disease development as well as the items that best describe these two autism spectrum disorder presentations. Maternal and infant data as well as ADOS-2 score were included in different ML testing models. Depending on the outcome to be estimated, a best-performing model was selected. RIDGE regression model showed that the best predictors for ADOS social affect score were gut disturbances, EEG retrievals, and sleep problems. Linear Regression Model showed that term pregnancy, psychomotor development status, and gut disturbances were predicting at best for the ADOS Repetitive and Restricted Behavior score. The LASSO regression model showed that EEG retrievals, sleep disturbances, age at diagnosis, term pregnancy, weight at birth, gut disturbances, and neurological findings were the best predictors for the overall ADOS score. The CART classification and regression model showed that age at diagnosis and weight at birth best discriminate between ASD and MSDD.

## 1. Introduction

Autism spectrum disorder (ASD) is a neurodevelopmental disorder characterized by impairment in communication and social interaction as well as repetitive, stereotyped behavior [1,2]. Since the publication of the Diagnostic and Statistical Manual of Mental Disorders, Fifth edition (DSM-5), ASD has mirrored an umbrella of diagnostic entities that previously reflected multiple distinct disorders, including Autistic Disorder, Asperger’s Syndrome, Multisystem Developmental Disorder (MSDD), and other clinical features previously defined as “pervasive developmental disorders” (PDD) [3].

To be diagnosed with Autism Spectrum Disorder, children must have:Deficits in social interaction and communication spanning all three of the following areas: (1) social-emotional reciprocity, (2) nonverbal communication, and (3) development, management, and understanding of interpersonal relationships;Stereotypic behavior, either motor or vocal/verbal, fixed patterns of behavior or strict adherence to routines, restricted interests and abnormal sensory processing;Autism onset in early childhood.

DSM-5 distinguishes three levels of autism severity based on social functioning, ranging from Level 1 (“requiring support”) to Level 3 (“requiring very substantial support”).

We can distinguish four main developmental trajectories at autism onset:Lack of acquisition of new developmental functions, especially expressive language and symbolic play, usually between 9 and 24 months;Loss of previously acquired functions (“regression”), typically between 12 and 30 months;Loss of previously acquired functions after 3 years of age (“late regression”);ASD in the context of a global developmental delay [4].

MSDD (DC: 0–3, diagnostic manual) is a term coined by Dr. Stanley Greenspan to diagnose children under the age of 3 who exhibit signs of impaired communication as in autism, but with stronger emotional attachment disturbances than in autism. MSDD is considered a milder developmental disorder than autism spectrum disorder, but the differences are very slight, and some argue that the differences lie “in the eyes of the clinicians” [5]. Nevertheless, children with MSDD who receive adequate treatment generally have better prognosis than most children diagnosed with ASD. Some clinicians use MSDD as a “temporary” diagnosis for children under the age of 3, when there are reasons that suggest to differ the diagnosis of autism (DC, 0–3) [6].

The definition of Multisystem Developmental Disorder in DC 0–3 [7] includes the following:Significant impairment, but not complete lack, of the ability to engage in an emotional and social relationship with a primary caregiver;Significant impairment in forming, maintaining, and/or developing communication. This includes preverbal gestural communication as well as verbal and nonverbal symbolic communication;Significant dysfunction in auditory processing (i.e., perception and comprehension);Significant dysfunction in the processing of other sensations, including hyper- and hyporeactivity (e.g., to visual–spatial, tactile, olfactory, proprioceptive, and vestibular input) and motor planning (e.g., sequencing movements).

Children up to three years of age with various language and communication disorders associated with relational and emotional problems may not fully fit the ASD clinical features and/or present non-fixed clinical pictures with gradual and favorable changes over time.

Multisystem Developmental Disorders therefore represent a concept referred to as a permanent and relatively fixed deficit but with openness to change and growth (DC, 0–3) [7].

For a young child whose development is rapid, changeable, and potentially flexible, it may be important to offer diagnostic alternatives.

This type of diagnosis would be taken into consideration when the ability to enter into relationships with others and to possess some of the prerequisites of communication in the area of intersubjectivity are observed.

Other than the clinical evaluation, the psychological assessment is useful to describe clinical features of ASD and other neurodevelopmental disorders. Psychological assessment is crucial for quantifying functional deficits and following their evolution over time, mapping not only the weaknesses but also the strengths [8].

The identification and definition of the individual functioning profile requires the evaluation of the intelligence quotient, the severity of autistic symptoms, and the adaptive quotient.

The assessment of the intellectual profile is usually performed by Leiter-3, a non-verbal scale that also excludes calculation, working memory and processing speed from the intellectual quotient (IQ). It emphasizes fluid intelligence and is a reliable assessment measure, without any cultural or language bias [9]. However, both the Leiter-3 and other psychometric tools may underestimate the intellectual potential of ASD subjects [10] due to the low ability of these children to adapt to the assessment setting. A more reliable instrument in describing the developmental profile of ASD children, from birth to eight years, is the Griffiths Mental Development Scale (GMDS) [11,12].

The Autism Diagnostic Observation Schedule—Second Edition (ADOS-2) is currently considered the “gold standard” in the assessment of ASD. It represents the best-practice clinical tool for the diagnosis of ASD in children [13] providing two different cut-off scores for “autism spectrum” and for “autism”. The first shows high specificity (76–86%) and acceptable sensitivity (81–94%), the second shows high sensitivity (95%+) and low specificity (63–73%) [14]. Although widely used in research because of its standardization, the ADOS requires clinical judgment from a professional with expertise in ASD when making a diagnosis. Despite that, the presence of false positives and the low predictability of ASD classification remain a challenge for clinicians.

Another useful tool for clinical evaluation of ASD is the Childhood Autism Rating Scale (CARS) that was developed to recognize young children with ASD symptoms and distinguish their severity in responses including smell, touch, light, sound, body use, social behavior, verbal and nonverbal communication, and coherence of intellectual response [15,16]. It can be considered an integrative diagnostic tool [17].

Despite having a high sensitivity for the presence/absence of ASD-typical signs, the above-mentioned diagnostic tools have an unavoidable degree of interrater unreliability among clinical users. New technologies can help the clinicians in the screening of the most effective tools allowing a more objective diagnosis of ASD in order to start a therapeutic intervention as early as possible. Among them, Artificial Intelligence-based methods have gained importance in the last years in light of their development, acceptance by the scientific and clinical community, as well as increasing trustworthiness. Among them, Machine learning (ML) methods have been used to support clinicians in solving complex tasks dealing with differential diagnosis, the clinical characteristics of ASD being somewhat similar to those of other neurodevelopmental and neuropsychiatric disorders. Dating back to 2016, one of the first attempts in this regard was made by Duda and colleagues [18] who employed ML models to select a subgroup of Social Responsiveness Scale (SRS) items to distinguish ASD from ADHD, highlighting the positive role for this innovative way to conduct such an investigation. This has fostered new related works, either aiming at performing a feature selection in terms of cognitive or mood measures to distinguish between clinical and non-clinical groups [19] or, closer to our aim, to detect ASD predictive features among adolescents and adults by picking up the most significant features from the ADOS in this regard [20].

Considering the success in terms of feasibility and positive results obtained by the works mentioned above, and in particular by the article by Küpper and collaborators, the aim of this study was to evaluate the ability of different ML models to improve discriminative power of the diagnostic procedure of ASD on the basis of maternal and infant data as well as the ADOS score. Therefore, differently from what is already published in the literature, here, we employed clinical and diagnostic features to assess the best predictors of disease development as well as the items that best describe the different autism spectrum disorder presentations. By retrospective data collection from patients with ASD and MSDD, this study aims to use the ML models to better describe ASD and other clinical conditions such as MSDD and clinical profiles. This represents a more difficult task than the performance of a differential diagnosis between ASD and non-clinical disturbances. This could potentially serve as a future reference study in this domain to identify early risk factors and clinical predictors for ASD with respect to other clinical disorders with somewhat similar features.

## 2. Materials and Methods

### 2.1. Study Cohort

Fifty-seven children (50 M, 7 F; age 3.5 ± 2.7 years) were recruited at the Unit of Child Neuropsychiatry of Polyclinic Hospital University in Messina (Italy) based on (1) fulfilling diagnostic criteria for neurodevelopmental disorders, including autism spectrum disorder (ASD) (38 M, 6 F; age 4 years, 22–144 months) and multi-systemic developmental disorder (MSDD) (12 M, 1 F; age 2.4 years, 19–36 months).

The diagnosis of ASD was confirmed by expert child psychiatrists at the time of inclusion.

An accurate anamnestic collection was carried out with the parents to investigate possible sleep or eating disorders, behavioral problems, alterations in electroencephalographic (EEG) activity, or other medical problems.

At the time of recruitment, Intellectual Quotient (IQ) or Developmental Quotient (DQ) were determined using the Leiter International Performance Scale—Third Edition [9] or the Griffith Mental Development Scales (GMDS) [11], respectively. Autistic behaviors were assessed using the ADOS-2 [9] and the Children Autism Rating Scales (CARS) [15]. Adaptive functioning was assessed using the Vineland Adaptive Behavior Scales—Second Edition [21].

The GMDS assesses a child’s strengths and weaknesses in all developmental areas, including the locomotor abilities (gross motor skills, balance, coordination, and movement control), personal–social capacities (independence in daily activities and in interaction with other children), hearing and language development (hearing, expressive language and receptive language), eye and hand coordination (fine motor skills, manual dexterity and visual monitoring skill). Furthermore, the developing ability to reason through tasks (including speed of working and precision) and the practical reasoning (ability to solve practical problems, understand basic math concepts and moral issues) can be evaluated. Three sub-quotients (personal–social, hearing and language, and practical reasoning) indicate the severity of symptoms when the diagnosis of ASD is assigned [12].

ADOS is a is a semi-structured, activity-based assessment aimed to evaluate communication skills, social interaction, and imaginative use of materials in individuals who are suspected to have ASD. It provides the examiner with the opportunity to observe behaviors directly relevant to the diagnosis of ASD. The ADOS-2 can be used for individuals regardless of age (from 12 months of age through adulthood), developmental level, and language skills because it comprises several modules. The comparison scores (from 1 to 10) indicate different levels of impairment: 1 indicates minimal-to-no evidence of autism-related symptoms and 10 indicates a very high level of autism-related symptoms.

### 2.2. Machine Learning Approach

The database included in this investigation was composed of data from 57 individuals diagnosed with neurodevelopmental disorders at an age of 12–144 months, including ASD and MSDD.

The tasks assigned to the machine learning (ML) algorithms included the predictions of clinical scores normally employed to categorize the severity of the existing condition, including ADOS and some of its subscales, and the Developmental Quotient (DQ). In addition, the prediction of differential diagnosis between ASD and MSDD was attempted. Since the dataset considered also includes the real values of the outcome to be predicted, the models employed in the present work are supervised ones, allowing for a so-called “task-driven approach”, where the aim, based on the input values, is to perform an estimation of the true value in output.

Machine learning models, where possible, were compared featuring the same parameters in order to allow a fair assessment of their respective performances.

The entire database was divided into a training and a test set based on an 80–20% split, which is a quite common choice in this regard. All the algorithms evaluated were considered taking into account a 10-fold cross-validation for performance assessment, which is popular to somewhat guarantee a reasonable decrease in terms of overfitting (i.e., the tendency of a model to fully adapt to the training set, without the capability to generalize over another dataset, including the test set, which is quite frequent especially in the datasets featuring a reduced number of observations, i.e., individuals). In order to further reduce the overfitting likelihood, the best model was considered to be not the one providing the best performances on the training set overall, but the one within one standard error from the best performances. This choice was adopted for all the models featuring regression tasks (i.e., the prediction of ADOS and subscales, as well as DQ), whereas the overall best model was picked up in the case of classification tasks (i.e., the differential diagnosis between ASD and MSDD) due to its intrinsically different characteristics. It should be specified that the best performances were evaluated in terms of minimal Root Mean Squared Error (RMSE) for the regression tasks and in terms of maximum accuracy for the classification tasks. In both cases, the error was considered in terms of the difference between the predicted value from a model and the real outcome value.

### 2.3. ML Models

#### 2.3.1. LASSO

The Least Absolute Shrinkage and Selection Operator, LASSO, is a very common ML model relying on a regression analysis method. It carries out both variable selection and regularization and aims at improving the prediction accuracy and the resulting model interpretability. It is known to be particularly useful when datasets are composed of several variables hypothesized not being useful for prediction purposes [22]. Here, it was employed for regression tasks.

#### 2.3.2. RIDGE

Ridge Regression is a ML technique often employed when the regression data to be analyzed are significantly affected by multicollinearity problems. If multicollinearity occurs, it turns out that least square estimates are totally unbiased with a large variance deviating them significantly from their true value. By adding a quota of bias to the regression estimates, RIDGE regression is able to reduce the standard errors. Contrary to LASSO, which is quite similar in some instances, RIDGE regression shrinks all the coefficients to a non-zero value. Here, it was employed for regression tasks [23].

#### 2.3.3. Elastic Net

The Elastic Net attempts at taking the advantages of both LASSO and RIDGE, blending their optimal characteristics. Its main regularization parameter can be continuously varied between 0 and 1, with the lower limit (zero) making the model equal to RIDGE and the upper limit (one) to LASSO. A 0.5 value indicates a 50/50 blend between the two regression models. Like the two above-mentioned models, the Elastic Net was also employed for regression tasks [24].

#### 2.3.4. CART

Classification and Regression Trees (CART) are popular and powerful ML models relying on the deconstruction of the overall sample into smaller groups performed through repeated, binary splits of the sample, considering one exploratory variable at a time. Their advantages are manifold: they can be easily adapted to different data, including cross sectional, longitudinal, survival data, the possibility to use different types of response variables, and the fact that they do not need to make any assumptions in terms of the normality of the data distribution. On the other hand, their main limitations include their strong sensitivity to data changes and their somewhat limited interpretability [25]. Due to its adaptability to both conditions, in the present study, it was employed for both regression and classification tasks.

### 2.4. Random Forest

Random Forest (RF) are learning methods that can be applied for classification and regression purposes, as in the present article, operating by building up a series (forest) of decision trees at the training. Their output is represented by the class that is the mode of the classes for classification, or the mean prediction for regression, of the individual trees. With respect to the classical decision trees, RF possess several advantages. Those include the performance of implicit on-the-run feature selection, the provision of accurate indicators of feature importance, the absence of the need for particular data preparation prior to the application of the ML model, and the opportunity for them to handle binary, categorical, numerical features without any need for scaling, normalization or standardization. They are also unlikely to perform overfitting and they are relatively quick to train and versatile, although their interpretability is often cumbersome [26].

### 2.5. Linear Model

A Linear Model was here employed for regression purposes, and, more specifically to this extent, a Linear Regression Model was selected. In a Linear Regression Model, the response variable “y” (target, the “outcome”) is expressed as a linear function or linear combination of all the predictors “X” (the observed variables). The underlying relationship between the response and the predictors is deemed to be linear (i.e., lying on a straight line), and the error distribution of the response variable should be normally distributed [27].

### 2.6. Neural Network

An Artificial Neural Network is a quite popular ML approach making use of artificial neurons connecting each other between nodes and weights between their connections under the principles of resemblance to biological neurons. The inputs to the network are modified step by step through weights and summed to each other, eventually triggering a related activation function. As such, various kinds of Neural Networks exist, among which for the present study a resilient backpropagation-based network, composed of two hidden layers of 10 and 4 neurons, respectively, was employed since it provided the best performances in terms of regression (and classification) with respect to other models tested, with a reasonably low computational burden [28]. An example of the topology used is shown in Figure 1.

All the models adopted were implemented and trained using an RStudio v.1.4.1106, running on a PC equipped with Intel(R) Pentium(R) CPU B980 @ 2.40 GHz featuring 8 GB RAM. Comparisons between the models were performed based on such technological basis.

## 3. Results

### 3.1. Clinical Features of the Study Population

The clinical features of children recruited for this study are shown in Table 1.

Time range for diagnosis of ASD or MSDD was 12–144 months in our study cohort. Age at diagnosis was lower than 36 months in 47.3% of the examined cases.

Almost half of the pregnancies were complicated by threats of miscarriage, uterine contractions, placental abruption, emergency caesarean section, gestational diabetes, intrauterine growth retardation, uterine fibroid, antibiotic administration, thromboflebitis, cardioaspirin treatment, and risk of retinal detachment (3.8%).

About 18% of births occurred between 31 and 37 weeks of gestation; moreover, birth occurred by caesarean section in more than two thirds of all cases.

About 14% of the recruited children were born with low birth weight (<2500 g), 42% of which were pre-term newborns.

In pre-term newborns, perinatal problems were encountered, including respiratory distress, hospitalization in Neonatal Intensive Care Unit, pulmonary hypertension and neonatal sepsis, corpus callosum agenesis, jaundice, hypotonia, and umbilical cord wrapped around the neck.

More than half of the examined cases exhibited development retardation, particularly denoted by a delay in language skills in all subjects.

About two fifths of ASD cases presented with comorbidities, including hyperactivity, epilepsy, anxiety disorder, ADHD, hyperkinesia, Tourette syndrome, Riddle syndrome, coeliac disease, microcephaly and severe intellectual disability, interatrial septal defect, and electroencephalogram abnormalities.

About two fifths of the children experienced either difficulties in falling asleep or frequent nighttime awakenings. Gut disturbances, including diarrhea, constipation, and encopresis occurred in one fifth of the children; one of the recruited children presented with allergies.

About 40% of subjects showed abnormalities in the EEG records.

About 60% of the examined cases had a positive family history for neuropsychiatric disorders.

### 3.2. ML Analysis

As mentioned, two kinds of tasks were assigned to the ML models implemented, namely regression and classification.

Regression tasks were applied when trying to predict continuous-like output variables, including the ADOS score and its subscores (e.g., Social Affect, Restricted and Repetitive Behaviors), as well as the DQ score.

For these predictions, as previously reported, the RMSE was selected as the parameter to be kept at the minimum to ensure the good performances of the model. Table 2 shows the RMSE values for the different models concerning the ADOS Social Affect subscore.

As such, the best performances, those with the lowest RMSE, were achieved by the RIDGE model using a hyperparameter lambda of 0.9 and capable of performing such prediction within 390.83 s employing the technological infrastructure described above.

The RIDGE model makes use of all the features with different “penalization” parameters, the largest of which, as absolute values, were assigned to gut disturbances, EEG retrievals, and sleep problems.

The other ADOS subscale evaluated was about the Restricted and Repetitive Behaviors, on which the Linear Model performed best (Table 3).

Regarding the clinical interest, the most reliable predictors for this specific task included the occurrence of term pregnancy, the psychomotor development status, and the occurrence of gut disturbances.

Of particular clinical interest is the prediction of the overall ADOS score, where the LASSO made the best use of the data available, outperforming all the other models as displayed in Table 4.

The hyperparameter used by the best, up to one Standard Error, LASSO model was 0.197 (see Figure 2), and the full model was trained within a very short amount of time, 32.24 s.

The LASSO model used seven parameters including the EEG retrievals, the occurrence of sleep disturbances, the age of neurodevelopmental disorder diagnosis, the occurrence of term pregnancy, the weight at birth, the occurrence of gut disturbances, and the retrievals from the neurological examination.

As reported above, the last regression task demanded was to predict the DQ score, with the CART performing at best (Table 5).

The CART model was trained within 121.6 s, and the best performances were achieved with a hyperparameter of 0.1 (Figure 3), making the best use of the age at diagnosis and the psychomotor development status.

On the other hand, the classification task was conducted to distinguish the subjects included in the database according to their clinical status, taking into account also the differences between ASD and MSDD. As reported above, three models were employed to this extent: CART, RF and Artificial Neural Network. According to the best accuracy data, CART and RF performed equally in the demanded task, with an 80% incidence of correct classification into the two classes. As expected, CART obtained such results in a shorter amount of time and using less computational resources (88.55 s for training) than RF did due to the lower complexity of the model itself. As such, the CART hyperparameter providing the best results was 0.9, and the most important variables for the specific task resulted to be the age at diagnosis and the weight at birth.

## 4. Discussion

ML is a subfield of Artificial Intelligence which has the potential to substantially enhance the role of computational methods in neuroscience. These machine learning models are based on statistical algorithms and are suitable for complex problems involving combinatorial explosions of possibilities or non-linear processes where traditional computational models fail in terms of quality or scalability [15].

In clinical studies, the prediction of a diagnosis or correlations between clinical variables is becoming a fundamental step. In these cases, ML is used, and it often enables superior opportunities compared to standard statistical approaches [16].

The determination of the clinical diagnosis, the establishment of the severity of the disorder, as well as the differential diagnosis between disorders appear to be difficult and complex tasks for a clinician. To this extent, ML can support clinicians in various domains. Those might include, among many others, the prediction of the clinical outcome (as the score reported in a given clinical questionnaire or scale) based on observational behavioral or instrumental variables, or the investigation around the possible relationships between different variables that might stay hidden when just the classical statistical approaches are employed.

Focusing on the present article, several supervised ML approaches have been applied and compared to the prediction of the typical clinical scores and variables commonly employed in the diagnostic pathway of the ASD and related conditions. In particular, the ADOS scores were predicted in the cohort of participants by means of different ML models, trying to understand which of the variables, taken into account in the clinical assessment, are deemed more predictive for the related outcome. The overall ADOS score was best predicted using a subset of seven variables: (a) the EEG retrievals, (b) the occurrence of sleep disturbances, (c) the age of neurodevelopmental disorder diagnosis, (d) the occurrence of term pregnancy, (e) the weight at birth, (f) the occurrence of gut disturbances, (g) the retrievals from the neurological examination.

Furthermore, the ADOS subscale Social Affect score was best predicted by combining the occurrence of gut disturbances, the EEG retrievals, and sleep problems, also significant in the prediction of the ADOS score, whereas the ADOS subscale Restricted and Repetitive Behaviors score was mainly related to the occurrence of term pregnancy, the psychomotor development status, and the occurrence of gut disturbances. Notably, two of these variables (term pregnancy and gut disturbances) were also seen in the previous analyses.

Taken together, these data make the assessment of gut disturbances particularly useful to check for the clinical variations featured in the ADOS clinical scores. This may suggest a pivotal role of the gut–brain axis and the microbiota in ASD and other brain disorders [29], as well as all neurodevelopmental disorders [30]. Several studies have successfully demonstrated the impact of the gut–brain axis on the fetal programming of brain development and the modulatory ability of various gut microbial metabolites in this process. Gut microbial metabolites have been shown to influence the release and turnover of neurotransmitters, the impairment and recovery pathways of synaptic plasticity, the alterations of parasympathetic activity, and the expression profiles of canonical signaling pathway [31].

In recent years, there has been an emerging interest in the possible role of the gut microbiota as a co-factor in the development of ASD as many studies have highlighted the bidirectional communication between the gut and the brain (gut–brain axis). Autistic children often carry on an intestinal dysbiosis which can exacerbate both CNS and GI malfunctioning in several ways. A skewed ASD-associated gut microbiota can boost the production of inflammatory cytokines, increasing intestinal permeability, and can negatively influence mRNA splicing in the CNS, yielding ASD-like behaviors in rodent models [32].

Concomitant medical symptoms, including gastrointestinal and sleep problems, are common in many children with ASD. In a study conducted in 2018, ASD children with gastrointestinal symptoms reported more severe underlying ASD symptoms than others. ASD children with sleep disorders performed lower in daily living skills, social cognition, social communication, and intellectual development than ASD children without sleep disorders [33]. This could mean higher scores on the ADOS assessment for this group of autistic patients.

The prediction of the DQ, on the other hand, was performed at best using as input variables the age at diagnosis and the psychomotor development status. This fact points out the importance of a timely diagnosis for such neurodevelopmental disorders and of a proper patient care in their early infancy or toddlerhood to improve their clinical outcome, reducing severe signs of the related clinical condition. From the ML perspective, our work somewhat confirms the findings in the literature, highlighting the added value of the ML approach with respect to linear models for a better prediction of DQ in our case, or of intellectual disabilities, as reported [34].

In the initial stages of clinical assistance, the evaluation of the developmental profile and cognitive abilities in children with ASD assumes a key role, allowing to establish the functioning profile. Early diagnosis and assessment of the level of development by determining the Development Quotient (DQ) and of cognitive abilities by measuring the Intelligence Quotient (IQ) are crucial aspects not only in delineating the clinical phenotype, and the long-term outcomes of subjects with ASD, but also in establishing early rehabilitation therapy [35].

The data provided by the machine learning approach can steer the diagnostic process in specific directions, shortening the time to diagnosis. The evidence of a specific pattern of symptoms and conditions, such as the occurrence of sleep and gut disturbances, allows to screen children at risk predicting a negative development trajectory and, thus, to start treatment earlier.

While current treatments vary, most interventions focus on managing behavior and improving social and communication skills to enable optimal social functioning and independence [36]. Our work in this framework represents a useful add-on to the current literature, which has progressed a lot since a few years ago, but is still affected by huge heterogeneity of data and retrievals [37].

Research examining the age of ASD diagnosis and the factors affecting it highlighted that the identification of factors associated with the age of ASD diagnosis is essential for a full understanding of the obstacles to early developmental screening and comprehensive diagnostic evaluations [38]. The last ML model attempted at performing a differential diagnosis between PDD and MSDD. This approach, making use of ML for purposes of differential diagnosis based on best-practice diagnostic instruments for autism, was already proposed with satisfying results, even though it was limited to a lower number of models from the ML perspective [39], focused on some specific clinical variables [40] or both [41]. In this respect, our work comparing different classification models showed that to the benefit of the clinician, the age at diagnosis and birth weight were seen to be the variables, which were the most predictive for distinguishing the two disorders. Recent meta-analysis studies showed that birth weight, especially low and very low birth weight, are associated with increased risk of ASD diagnosis [42]. Over the recent years, after the publication of the DSM5^3^, the interest in MSDD has declined. The nosographic autonomy of MSDD in now questionable due to several reasons: (a) the increased use of “spectrum” dimension; (b) the introduction of “hyper- or hyporeactivity to sensory input or unusual interest in sensory aspects” in the ASD diagnostic criterion “B”; (c) the DSM5 statement that ASD symptoms may be seen even earlier than 12 months if developmental delay is severe However, prompt recognition of the MSDD clinical feature still appears to be of great importance because children who are affected share a developmental trajectory in continuity with ASD, with most symptoms and risk factors shared with ASD children. According to our data, MSDD could represent a sort of “bridge” between a high-risk predisposition to autism and the full autism spectrum disorder diagnosis. The shared profile of clinical variables and assessment scores between MSDD and ASD shown in our research is consistent with the notion that a fully severe autism disorder can be developed independently of the severity of the initial evaluation.

## 5. Conclusions

ASD has a multifactorial etiology, making it difficult to acquire timely and accurate diagnoses. Nevertheless, ASD diagnosis is currently earlier and more accurate due to increased public awareness and widespread research in the field of autism. In most cases, symptoms are clearly evident as early as 18 months of age, reducing the learning opportunities and affecting the development of the social, cognitive and behavioral competences in a child. The value of early diagnosis in patient clinical outcome, is now clear, as well as the fact that evidence-based interventions can significantly improve the quality of life of people with ASD and their families.

Through better and earlier identification, it will be possible to promptly recognize children at risk and implement therapeutic interventions aimed to significantly reduce the impact of this disorder on the global development of the child. It can be hypothesized that in the future, timely help to the child and their parents based on the early identification of risky situations could prevent the development of this serious disorder of mental health, reducing the frequency of its occurrence.

Despite the methodological limitations, including the quite different age ranges between the two clinical groups and the relative simplicity of the ML models employed here, the present work shows that the application of ML methods to the clinical data collected in patients with neurodevelopmental disorders results in a better understanding and prediction of clinical scores useful for detection of ASD and classification of disease severity.

Examining the ongoing research in the field of neurodevelopmental disorders, it is likely that, as is the case in other clinical conditions, the transfer of knowledge to ML models and the implementation of the same ML models will provide specialists with predictive and explainable models for more accurate ASD detection and differential diagnosis.

## Figures and Tables

**Figure 1 brainsci-13-00883-f001:**
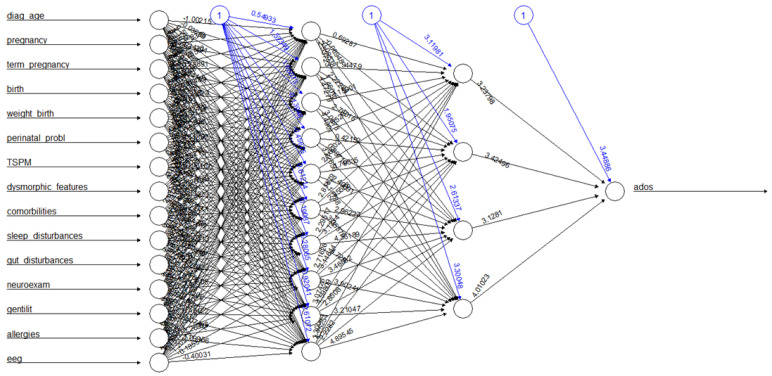
A topology example of a Neural Network employed in the present work.

**Figure 2 brainsci-13-00883-f002:**
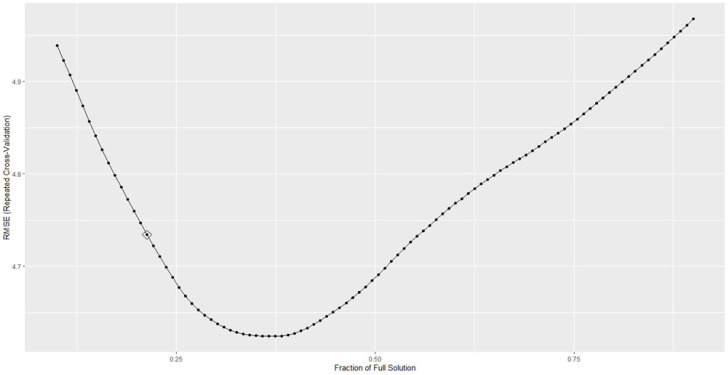
RMSE obtained by the LASSO model on predicting ADOS varying with the hyperparameters.

**Figure 3 brainsci-13-00883-f003:**
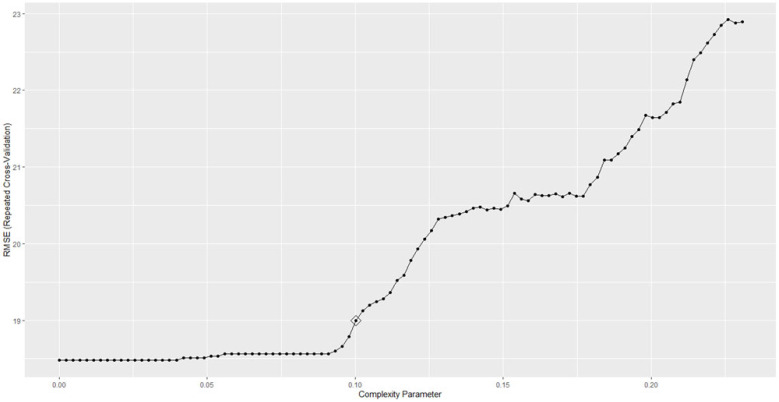
RMSE obtained by the CART model on predicting DQ varying with the hyperparameters.

**Table 1 brainsci-13-00883-t001:** Clinical features of 57 patients recruited for this study.

Clinical Features		Number of Cases (%)
Age at diagnosis (months)		41.5 ± 27.9
Physiological pregnancy	Yes	31 (54.4%)
	No	26 (45.6%)
Term pregnancy	Yes	47 (82.4%)
	No	10 (17.6%)
Birth	Eutocic delivery	16 (28.05%)
	Caesarean section	40 (70.2%)
	Dystocial	1 (1.75%)
Weight at birth (kg)		3.213 ± 0.957
Dysmorphic features	Yes	2 (3.5%)
	No	55 (96.5%
Perinatal problems	Yes	10 (17.6%)
	No	47 (82.4%)
Psychomotor development	Developmental delay	30 (52.6%)
	Typical development	27 (47.4%)
Comorbidities	Yes	21 (36.8%)
	No	36 (63.2%)
Sleep disturbances	Yes	22 (38.6%)
	No	35 (61.4%)
Gut disturbances	Yes	24 (42.1%)
	No	33 (57.9%)
Neuroexamination	Normal	42 (73.7%)
	Higher than normal	15 (26.3%)
Gentilitium	Positive	33 (57.9%)
(Family history)	Negative	24 (42.1%)
Allergies	Yes	1 (1.8%)
	No	56 (98.2%)
EEG	Normal	36 (63.1%)
	Higher than normal	21 (36.9%)
ADOS score		16.3 ± 4.95
ADOS Social Affect subscore		12.8 ± 3.9
ADOS Repetitive and Restrictive Behavior subscore		3.5 ± 1.7
Developmental Quotient		61.2 ± 19.9

Legend: EEG: electroencephalography; ADOS: Autism Diagnostic Observation Schedule.

**Table 2 brainsci-13-00883-t002:** Performances of the different models trained on the ADOS Social Affect score prediction.

Model	RMSE
Linear Model	4.514
Neural Network	5.126
LASSO	3.859
RIDGE	3.589
Elastic Net	4.067
CART	3.759
Random Forest	3.926

**Table 3 brainsci-13-00883-t003:** Performances of the different models trained on the ADOS Restricted and Repetitive Behaviors score prediction.

Model	RMSE
Linear Model	1.311
Neural Network	2.526
LASSO	1.765
RIDGE	1.841
Elastic Net	1.789
CART	1.747
Random Forest	1.699

**Table 4 brainsci-13-00883-t004:** Performances of the different models trained on the ADOS score prediction.

Model	RMSE
Linear Model	4.947
Neural Network	7.533
LASSO	4.902
RIDGE	5.121
Elastic Net	4.920
CART	5.912
Random Forest	5.248

**Table 5 brainsci-13-00883-t005:** Performances of the different models trained on the DQ score prediction.

Model	RMSE
Linear Model	25.800
Neural Network	32.390
LASSO	22.915
RIDGE	25.429
Elastic Net	23.320
CART	17.221
Random Forest	21.666

## Data Availability

The datasets generated and analyzed during the current study are not publicly available due to medical confidentiality but are available from the corresponding author on reasonable request.

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
