# Peer review of "A Machine Learning Approach to the Diagnosis of Autism Spectrum Disorder and Multi-Systemic Developmental Disorder Based on Retrospective Data and ADOS-2 Score"

_brainsci, 2023, doi:10.3390/brainsci13060883_

Round 1
Reviewer 1 Report
Title: A machine learning approach for the differential diagnosis of autism spectrum disorder and multi-system developmental disorder based on retrospective data and ADOS-2 score
Abstract:
1. Please provide interpretations (e.g., Which modal had the best performance) and brief clinical implications at the end of the abstract.
Introduction:
1. In the introduction, please define Multisystem developmental disorder (MSDD). I suggest moving lines 327-336 to the introduction section to better introduce the disorder.
2. Please also provide clinical rationales for classifying ASD and MSDD. For example, will the differential diagnosis lead to a better choice of interventions?
3. The assessments, including Leiter-3 and GMDS, are very specific to the current study. I suggest moving them to the method section. Instead, please provide broader views of the developmental profile of children with ASD in the introduction.
4. There’s a lack of literature review about applying machine learning in the differential diagnosis of disorders in the introduction. Please summarize the findings from previous literature, suggest the research gap, and provide hypotheses specific to the aims of the study.
Methods:
1. Please include citations for the statements about each ML model.
2. The age ranges of ASD (22~144 months) and MSDD (19~36 months) are very different. This might affect the accuracy of using machine learning to classify the two diagnoses and their applications in clinical settings. Please include this as a study limitation.
3. The clinical features included in this article (e.g., sleep disturbance, gut disturbance, EEG, etc.) should be defined in the method. For example, how are they measured and what parameters (especially for EEG) were input to the modal?
Results and discussion:
1. Please provide study limitations and suggestions for future study at the end of the discussion section.
2. I suggest better describing the clinical implications and their rationale in the discussion sections. For example, since the current study used a sample that has already been diagnosed with ASD and MSDD, how machine learning could aid in the current diagnostic process and help with early identification? In addition, how better classification between ASD and MSDD would help in early intervention?
Author Response
please see the attachement

Reviewer 2 Report
-The discussion of all of the different instruments for assessing ASD is very detailed, but there is very limited information about the machine learning approach to diagnosis and the benefits it provides. More information in this regard is needed in the literature review, as well as any research that would support the use of machine learning for diagnosis.
-Please provide additional information about the methods used in this study. Please explain how machine learning actually works on a very fundamental level, as your readers may not all be well-versed in the technology.
-The descriptive statistics about the sample provided in the Results section, while interesting, don’t really provide a great of utility for actually making sense of what the study is trying to do. Might consider condensing in order to provide more information about the machine learning approach in the literature review and Methods section.
Round 2
Reviewer 2 Report
All comments appear to have been addressed